# Synthesis and Characterization of Magnetic Molecularly Imprinted Polymer for the Monitoring of Amoxicillin in Real Samples Using the Chromatographic Method

Rosario López [1], Sabir Khan [1,2,3], Sergio Espinoza Torres [1], Ademar Wong [2,4], Maria D. P. T. Sotomayor [2,4,*] and Gino Picasso [1]

1   Technology of Materials for Environmental Remediation Group (TecMARA), Faculty of Sciences, National University of Engineering, Av. Tupac Amaru 210, Rimac 15333, Peru
2   Institute of Chemistry, São Paulo State University (UNESP), Araraquara 14801-900, SP, Brazil
3   Department of Natural Sciences, Mathematics and Statistics, Federal Rural University of the Semi-Arid, Mossoró 59625-900, RN, Brazil
4   National Institute of Alternative Technologies for Detection, Toxicological Evaluation and Removal of Micropollutants and Radioactives (INCT-DATREM), Araraquara 14801-970, SP, Brazil
*   Correspondence: m.sotomayor@unesp.br

**Abstract:** Amoxicillin (AMX) is an antibiotic frequently used for the treatment of bacterial disorders and respiratory problems in both humans and animals. This work aims to synthesize a molecularly imprinted superparamagnetic polymer (SP-MIP) with a core-shell structure for the selective detection of AMX in real samples. Magnetite superparamagnetic nanoparticles (SNP) were prepared by the polyol method, coated with silica, and functionalized with silane groups. The polymerization process was executed using the free-radical precipitation method. Thermogravimetric analysis (TGA) was used to evaluate the thermal stability of the synthesized materials. The results obtained from $N_2$ adsorption and desorption analyses showed that the surface area of SP-MIP (19.8 $m^2/g$) was higher than that of the non-molecularly imprinted superparamagnetic polymer (SP-NIP—9.24 $m^2/g$). The optimized adsorption analysis showed that both SP-MIP and SP-NIP followed SIP-type behavior, with adsorption constant $K_S$ 0.01176, 1/n 1.73. The selectivity tests showed that SP-MIP is highly selective for AMX in the presence of other molecules. Finally, for the recovery analysis, the application of SP-MIP for determining AMX in samples of tap water, river water, and drugs using HPLC yielded a mean recovery value of 94.3%.

**Keywords:** amoxicillin; core shell; superparamagnetic nanoparticles; MIP

## 1. Introduction

Amoxicillin (AMX) belongs to the β-lactam group. It is one of the most important drugs used in clinics and hospitals worldwide and is widely used for the treatment of infectious diseases in both humans and animals [1]. AMX is constituted by a structure of variable side chains based on a β-lactam ring, which is responsible for antibacterial activity. After consumption, approximately 60% of the drug is excreted unchanged in the urine, and the drug residue ends up being disposed of in rivers, lakes, and other water bodies. The presence of AMX in these water bodies eventually leads to the contamination of humans and other living beings [2]. Its metabolites in the environment can lead to allergic reactions, antibiotic resistance, and the development of other related diseases. In this sense, the development of analytical methodologies that are capable of effectively helping monitor this compound in aqueous matrices is essential [3]. Several methods have been applied to determine the presence of AMX in aqueous matrices, including spectrophotometry [4–6], electrochemical methods [7,8], flow injection analysis [9,10], capillary electrophoresis [11,12], atomic absorption spectroscopy [13], and chromatography [14,15].

One of the relevant materials used in separation processes that has gained considerable traction among researchers in the last few years are magnetite nanoparticles (NPS-$Fe_3O_4$). This material has become increasingly popular due largely to its extraordinary magnetic and electrical properties [16], which allow it to be employed for a wide range of purposes, including bulk catalysts [17,18], catalytic support [19–21], core material in catalysts or core-shell materials [22–24], and electronic devices. Furthermore, owing to its high degree of stability and biocompatibility, NPS-$Fe_3O_4$ are also employed in different areas, including industries [25–27], medicine/pharmacology [28–30], biomedical applications [31,32], and in the environmental field [33–35]

Magnetite nanoparticles can be synthesized through different methods, including co-precipitation, hydrothermal, microemulsion, solvothermal, and polyol techniques. The application of these synthesis methods enables us to obtain magnetite nanoparticles of different sizes, shapes, dispersion, and crystalline structures with different properties. The synthesis of the SP-MIP was conducted using the polyol polymerization technique, as detailed by Cai and Wan [36], who used iron salt of acetylacetonate, Fe(acac)$_3$, as a precursor. Its decomposition takes place via polyols, such as triethylene glycol (TREG), that work simultaneously as a hydrophilic solvent for the iron salt at 120 °C as well as a reducing agent producing enough $Fe^{+2}$ for the formation of magnetite. Finally, due to its long polar terminations, it keeps the magnetite particles separated and prevents their agglomeration. Nanoparticles between 7–10 nm homogeneity are achieved, which makes them superparamagnetic and hydrophilic character.

The present study sought to determine AMX through the application of high-performance liquid chromatography (HPLC) using molecularly imprinted superparamagnetic polymer (SP-MIP), which was constructed using magnetite superparamagnetic nanoparticles (SNP) and the molecular imprinting technique. The SNP were obtained through applying tri ethylene glycol (TREG) as solvent.

The most important advantages of the proposed method are that it does not require any pretreatment procedures, such as filtration and extraction, and all the analyses are performed at room temperature. Following the optimization of the method with standard solutions, the technique was successfully applied to determine amoxicillin in different real samples.

## 2. Materials and Methodologies

### 2.1. Chemicals

Ethyl acetate (EtAcet, $CH_3COOC_2H_5$. ACS reagent, $\geq$99.5%), iron (III) acetylacetonate (Fe(acac)$_3$, $C_{15}H_{21}FeO_6$, for synthesis), ethanol absolute (EtOH, $CH_3CH_2OH$, for analysis EMPARTA® ACS), ammonia solution (NH$_4$OH, p.a. EMSURE® ACS, Reag. Ph Eur), toluene ($C_6H_5CH_3$, for analysis EMSURE® ACS, ISO, Reag. Ph Eur), tetraethyl orthosilicate (TEOS, Si(OC$_2$H$_5$)$_4$, for synthesis), methanol (MeOH, $CH_3OH$, for liquid chromatography LiChrosolv®), urea (NH$_2$CONH$_2$, ACS reagent, 99.0–100.5%) were purchased from Merck, Supelco, Milwaukee, Wisconsin, USA and amoxicillin (AMX, $C_{16}H_{19}N_3O_5S$, 95.0–102.0% anhydrous basis), acrylamide (Aam, $C_3H_5NO$, for synthesis), 3-(Methacryloyloxy)propyl] trimethoxysilane (MPS, H$_2$C=C(CH$_3$)CO$_2$(CH$_2$)$_3$Si(OCH$_3$)$_3$, $\geq$97%), potassium persulfate (KPS, $K_2S_2O_8$, ACS reagent, $\geq$99.0%), N,N′ -methylene(bisacrylamide) (MBAA, H$_2$C=CHCONH)$_2$CH$_2$, 99%), Caffeine (CAF, $C_8H_{10}N_4O_2$, powder, ReagentPlus®), ciprofloxacin (CIPRO, $C_{17}H_{18}FN_3O_3$, $\geq$98% (HPLC), uric acid ($C_5H_4N_4O_3$, $\geq$99%, crystalline) were purchased from Sigma Aldrich, USA. All the solutions were prepared using Milli-Direct Q®-3 ultrapure water of 18.2 M$\Omega$ cm$^{-1}$ at 298 K (Millipore).

The tap water sample was obtained from a laboratory in Lima, Peru, while the river water sample was obtained from a river located in the city of Puquio, Huacayo, Perú.

### 2.2. Computational Simulation

The preliminary study was conducted with a focus on theoretical determination analyses through a computational simulation in which the functional monomer effectively

interacted with AMX, where water was used as a solvent, leading to better results in the synthesis and selectivity of the MIP [37,38]. The semi-empirical simulations were performed using HyperChem® 8.0.5 (Hypercube, Inc., Gainesville, FL, USA). These simulations are necessary for designing the molecules since they serve as an input parameter for the other software employed in the study.

Table 1 shows the 20 monomers employed in the simulation analysis and the interaction of AMX. The OpenEye® software (OpenEye Scientific Software, Inc., Santa Fe, NM, USA), which performs several functions in the simulation, was also employed. This software comes with VIDA 3.0.0, used to visualize the molecules modeled with HyperChem and to verify possible bonding errors between the atoms and Omega2, which generates various conformers of the molecules. This helps when considering possible spatial impediments of a particular conformer. AutoIt 3.3.6.0 is a free BASIC-like scripting language designed to automate the Windows graphical interface and scripts in general. The simulation experiments were carried out based on the method applied for ciprofloxacin, described by Marestoni et al. [39].

**Table 1.** Monomers used in the computational simulation analyses for the most stable interaction between AMX and monomers.

| Sigla | Monomer |
| --- | --- |
| MP1 | N,N-methylenbisacrilamide |
| MP2 | Imidazole-4-acrylic acid |
| MP3 | Imidazole-4-acrylic ethyl ester |
| MP4 | Acrylic acid |
| MP5 | Acrylamide |
| MP6 | Acrolein |
| MP7 | Alylamine |
| MP8 | Acrylonitrile |
| MP9 | Ethylene glycol Dimethacrylate |
| MP10 | 2-(cyanoethyl amine)ethylmethacrylate |
| MP11 | Methylensuccinic acid |
| MP12 | Methacrylic acid |
| MP13 | 3-divinylbenzene |
| MP14 | 4-divinylbenzene |
| MP15 | Estiren |
| MP16 | 1-vinylimidazole |
| MP17 | 2-vinylpyridine |
| MP18 | 4-vinylpyridine |
| MP19 | 2-acrylamide-2-methyl-1-propanesulfonic acid |
| MP20 | 2-hydroxyethyl methacrylate |

*2.3. Synthesis of a Molecularly Imprinted Superparamagnetic Polymer with Core-Shell Structure (SP-MIP)*

A molecularly imprinted superparamagnetic polymer with a core-shell structure (SP-MIP) was prepared based on the procedure described by R. López [8]. First, magnetite SNP (Figure 1) were synthesized by the polyol method using Fe(acac)$_3$ and TREG initially under a controlled temperature of 120 °C for 30 min, and subsequently at 180 °C for an additional period of 30 min, and finally at 280 °C for 60 min [40,41]. The material under synthesis was coated with TEOS-based silica in ethanolic and ammoniacal media to chemically stabilize it (Fe$_3$O$_4$@SiO$_2$); this was followed by applying the Stöber sol-gel method [42]. Finally, the material's surface was functionalized with silane groups (Fe$_3$O$_4$@MPS) until the coating was effectively completed.

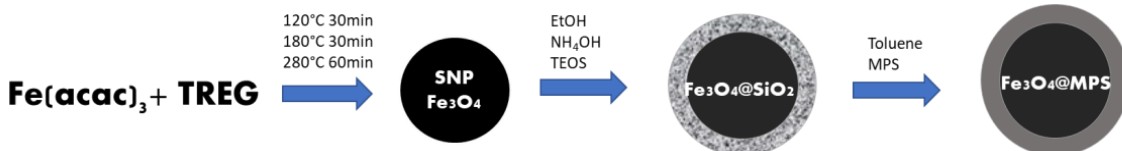

**Figure 1.** A schematic diagram illustrating the synthesis of the magnetite superparamagnetic nanoparticles (SNP) along with the stabilization and silanization processes.

Polymerization was carried out as a second step using the free-radical precipitation method for 3 h at 60 °C; the procedure was conducted using AMX (analyte), AAm (functional monomer), MBAA (crosslinker), and KPS (radical initiator) in the ratio of 1:4:100:0.185 mol, respectively. The polymerization process gave rise to SP-MIP. In addition, a non-molecularly superparamagnetic imprinted polymer (SP-NIP) was prepared using the aforementioned procedure but without adding AMX. Finally, AMX was removed from the previously synthesized polymer (SP-MIP). The extraction of AMX was performed using a Soxhlet extraction system with mixtures of methanol and water for 8 h (90:10, 70:30 $v/v$). High-performance liquid chromatography with ultraviolet (HPLC-UV) detection analysis was applied to ensure that all the AMX molecules (templates) had been removed from the supernatant. Finally, the SP-MIP and SP-NIP were repeatedly washed with water under a vacuum at 60 °C.

### 2.4. Characterization Experiments

Surface area, average pore sizes, and adsorption isotherms were determined by $N_2$ adsorption measurements at 77 K (liquid nitrogen temperature) using Micromeritics Gemini VII 2390. Both the SP-MIP and SP-NIP were subjected to a pretreatment analysis which consisted of degassing the polymers with helium at 80 °C for 4 h to remove any non-dissolvable or remaining substances on the surface of the materials. In addition, the Barrett-Joyner-Halenda (BJH) and Brunauer–Emmett–Teller (BET) methods were also used to calculate the surface area and average pore sizes of the polymers, respectively. Both the SP-MIP and SP-NIP (using about 10 mg of the polymers) were also subjected to thermogravimetric analyses (TGA) (about 10 mg) using a Thermal Analyzer STA6000 (PerkinElmer, Waltham, MA, US) under an inert atmosphere of nitrogen and by heating the samples from 40 to 800 °C with a heating rate of 20 °C min$^{-1}$. The FTIR–Vertex 70 Spectrometer Bruker Shimadzu (Anan, Japan) was utilized to analyze the various functional groups used to fabricate the materials. Finally, to examine the surface morphology of the synthesized nanomaterials, scanning electron microscopy (SEM) was performed with a JSM-7500F microscope.

### 2.5. High-Performance Liquid Chromatography Analysis of AMX

The HPLC analysis of AMX was performed at the Department of Analytical Chemistry of the Institute of Chemistry, São Paulo State University, Brazil, using a Shimadzu chromatograph 20A coupled to a UV-Vis SPD 20A detector with SIL-A autosampler and DGU-20A5 degasser controlled by a microcomputer. Column $C_{18}$ (250 mm × 4.6 mm) was used as the stationary phase, while methanol:water (40:60 $v/v$) at a flow rate of 1.0 mL min$^{-1}$ was used as the mobile phase. The sample injection volume and wavelength applied were 20 µL and 230 nm, respectively. AMX standards were prepared by diluting aliquots of 5, 10, 15, 20, 25, and 30 mg L$^{-1}$ of AMX from a 500 mg L$^{-1}$ stock solution (pH 7.0–7.2).

All the adsorption experiments were performed using 2.0 mL AMX solution in an aqueous medium. The SP-MIP was added to the solution, and the mixture was shaken using a rotary shaker. Subsequently, the magnetic SP-MIP was separated using an external neodymium magnet, while the supernatant solution was separated using a 0.22 µm PTFE membrane filter and analyzed by the chromatographic method. The study of the adsorption capacity of SP-MIP involves finding the optimal parameters of pH, AMX concentration,

amount of SP-MIP, and agitation time. The adsorption capacity $Q_e$ of the AMX in the SP-MIP was calculated using Equation (1):

$$Q_e = \frac{(C_0 - C_e) \times V}{m} \tag{1}$$

where $C_0$ (mg L$^{-1}$) is the initial AMX concentration, $Ce$ (mg L$^{-1}$) is the remaining AMX concentration, $V$ is the volume of the solution in mL, and m is the applied mass of SP-MIP or SP-NIP. AMX standards of 5, 10, 15, 20, and 25 mg L$^{-1}$ were also prepared to construct the calibration curve.

The results related to AMX adsorption in SP-MIP were obtained by plotting AMX concentration at equilibrium $Ce$, and the quantity of AMX adsorbed at equilibrium $Qe$ using isotherm models including Langmuir (Equation (2)), Freundlich (Equation (3)), and SIPS (Equation (4)). In Equations (2)–(4), which are related to the isotherm models, $K$ stands for the constant affinity adsorbent of each model, $Q_{max}$ is the maximum adsorption capacity of SP-MIP for AMX, and $\beta$ is the heterogeneity factor [43,44]:

$$Q_e = \frac{Q_{max} K_L C_e}{1 + K_L C_e} \tag{2}$$

$$Q_e = K_F C_e^{\beta} \tag{3}$$

$$Q_e = \frac{Q_{max} K_{SIPS} C_e^{\beta}}{1 + K_{SIPS} C_e^{\beta}} \tag{4}$$

*2.6. Selectivity for AMX Binding*

To evaluate the selectivity of SP-MIP and SP-NIP toward AMX, experiments were conducted using AMX in the presence of possible interfering molecules like urea, uric acid, caffeine, and ciprofloxacin. All the solutions of interferents were prepared similarly to prepare AMX standard solutions, and further experiments related to the analysis of the selectivity of the polymers toward AMX were carried out under the same optimal experimental conditions obtained in the adsorption study. For each interferent, standard solutions were prepared, and analytical curves were obtained using HPLC. The data obtained from these measurements were used to determine the distribution coefficient $K_d$ (mL/g, Equation (5)), the selectivity ($S$, Equation (6)) impression factor ($I_{factor}$, Equation (7)), and the relative selectivity coefficient ($K_{SR}$, Equation (8)); these variables were calculated based on the procedure described by Ndunda et al. [45] and using Equations (7)–(10) below:

$$K_d = \frac{Q_e}{C_e} = \frac{(C_0 - C_e) \times V}{C_e \times m} \tag{5}$$

$$S = \frac{K_{d(AMX)}}{K_{d(i)}} \tag{6}$$

$$I_{factor} = \frac{K_{d(SP-MIP\ AMX)}}{K_{d(SP-NIP\ AMX)}} \tag{7}$$

$$K_{SR} = \frac{I_{(factor,AMX)}}{I_{(factor,i)}} S \tag{8}$$

*2.7. Real Samples*

The performance of the synthesized SP-MIP was evaluated by testing two real samples: river water (collected from the Puquio river located in Huancayo, Peru) and tap water (obtained from the laboratory). Both samples were enriched with AMX of different

concentrations. An amount of 2 mL of the samples was spiked with a standard solution of AMX at three different concentration levels, and the pH was adjusted between 7.0 and 7.2 using NaOH (2 mol $L^{-1}$). Thereafter, 6 mg of SP-MIP was added to the mixture, which was shaken in a rotary shaker for 90 min. The SP-MIP polymers were separated using a neodymium magnet and were filtered through a PTFE membrane. Finally, the supernatant solution was analyzed by HPLC. This procedure was performed in triplicate.

## 3. Results and Discussion

### 3.1. Computational Simulation

The results obtained from the simulation analyses are presented in Figure 2; as can be clearly observed, the monomers M5 (acrylamide), M1 (N,N-methylene bisacrylamide), and M3 (Imidazole-4-acrylic ethyl ester) have higher binding energies and are found to be the best choices for application toward the synthesis of the polymers.

The monomers M5, M1, and M3 possess nitrogen atoms capable of producing a hydrogen bond with the amoxicillin molecule, leading to the formation of highly stable complexes, which effectively enhance the selectivity of the MIP. For the present study, the monomer acrylamide M5, with a binding energy of $-141.6$ kJ $mol^{-1}$, was chosen as a functional monomer.

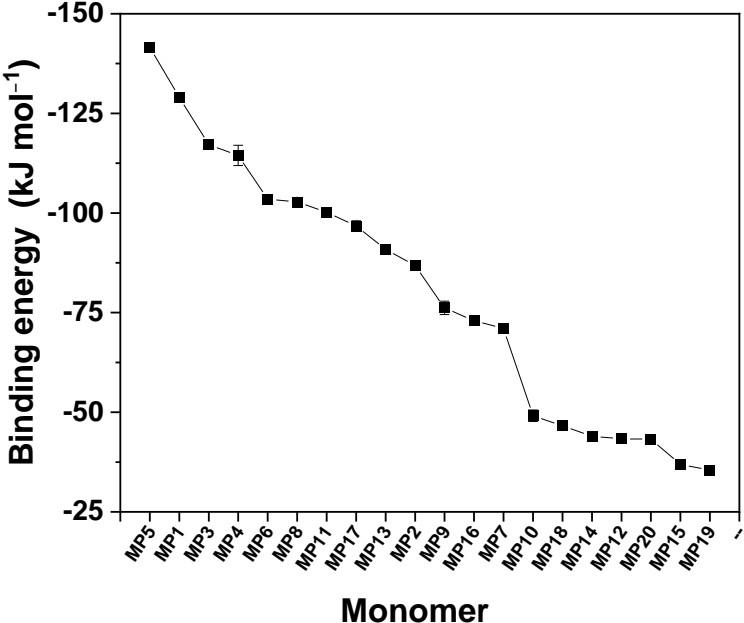

**Figure 2.** Monomers used in the semi-empirical computational simulations to analyze AMX in water.

### 3.2. Characterization Experiments

The specific surface area of SP-MIP and SP-NIP were evaluated through applying $N_2$ adsorption-desorption isotherm measurements. The isotherms (Figure 3) present wide and asymmetrical hysteresis loops. The results obtained from the BET analysis are presented in Table 2, where some differences between the SP-MIP and its respective SP-NIP can be highlighted. As can be noted, the specific surface area of SP-MIP (155.5 $m^2$ $g^{-1}$) was slightly higher than that of SP-NIP (109.3 $m^2$ $g^{-1}$); this shows that the SP-MIP adsorption level was somewhat higher than that of SP-NIP, and preliminary tests indicate that the polymers with core-shell structure generally have a greater surface area than those without this structure (SP-MIP: 19.8 $m^2$ $g^{-1}$; SP-NIP: 9.2 $m^2$ $g^{-1}$). This behavior may be attributed to the fact that the magnetite SNPs influence the coating process of the polymer, and this allows a more homogeneous deposition of the polymer along with a larger surface area. An analysis of the isotherms also showed that the synthesized polymers are mesoporous with almost similar average pore diameters.

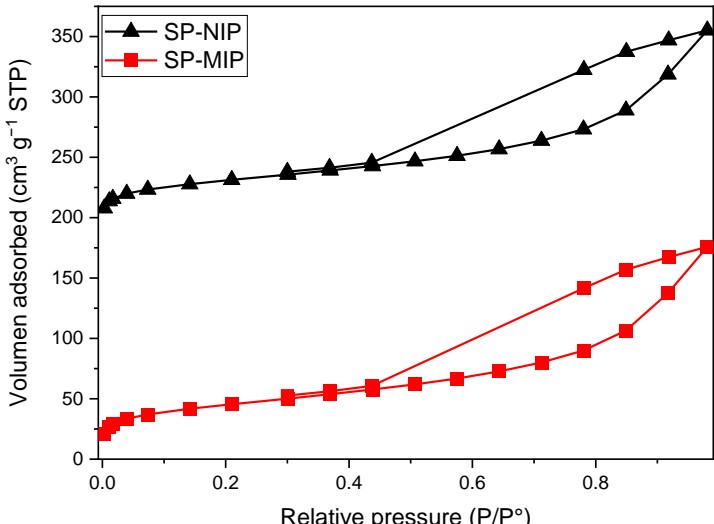

**Figure 3.** $N_2$ adsorption-desorption isotherms for superparamagnetic imprinted polymer (SP-MIP) and superparamagnetic non-imprinted polymer SP-NIP.

**Table 2.** BET surface area and porosity obtained for the SP-MIP and SP-NIP materials.

| Sample | BET Surface Area ($m^2$ $g^{-1}$) | Mesopore Area ($m^2$ $g^{-1}$) | Average Pore Diameter (nm) |
|--------|--------|--------|--------|
| SP-MIP | 155.5 | 129.5 | 8.5 |
| SP-NIP | 109.3 | 83.84 | 7.8 |

Figure 4 shows the TGA curves obtained for SP-MIP and SP-NIP, which are aimed at determining the response of these materials to temperature changes and calculating the content of the organic shell covering the surface of the core functionalized $Fe_3O_4$ [46]. Both the SP-MIP and SP-NIP exhibited similar behavior, and the total process of thermal decomposition occurred in two stages. The first weight loss (−15% SP-MIP and −25% SP-NIP) occurred in the temperature range of 30–140 °C; this was attributed to eliminating water physically adsorbed in the polymer. The second weight loss occurred in the temperature range of 320–500 °C; this loss (−50% in both) was attributed to the elimination of volatile substances and the combustion of organic compounds [47,48]. The assessment of the second weight loss leads us to the conclusion that only 50% of the mass of the SP-MIP and SP-NIP represents the adsorbent polymer, while the remaining 50% is the core (functionalized magnetite SNP).

Scanning electron microscopy (SEM) was used to examine the surface morphologies of the constructed material. Figure 5 shows that images from the micrographs indicated that SP-MIP was composed of sphere-shaped particles and had a greater porosity when compared with SP-NIP. Moreover, the surface of the produced SP-MIP has improved sites and a unique structure rougher texture, providing a greater surface area and porosity for the recognition of the target compound. In Figure 5a, we can see that the magnetite NPS has a diameter between 8 and 10 nm. In Figure 5b, the magnetite is protected and surrounded by a thick layer of $SiO_2$, in which it can be seen that the diameter has changed considerably. The particles, completely covered with TEOS and polymerized, reached a diameter of up to 300 nm, having the shape of the almost perfect sphere as discussed by the Stöber synthesis [42]. Figure 5c shows $Fe_3O_4@SiO_2$ modified with MPS that have almost maintained their spherical shape but have increased their size to approximately 500 nm.

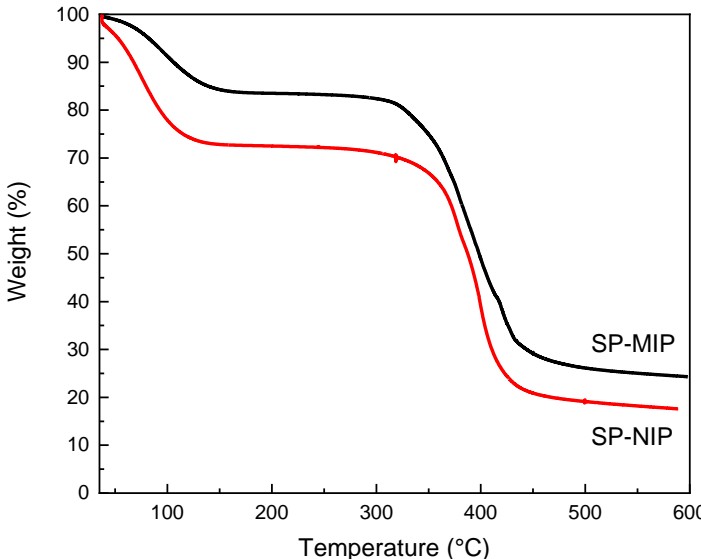

**Figure 4.** Thermogravimetric analysis (TGA) curves obtained for the superparamagnetic Imprinted polymer (SP-MIP) and superparamagnetic non-imprinted polymer SP-NIP.

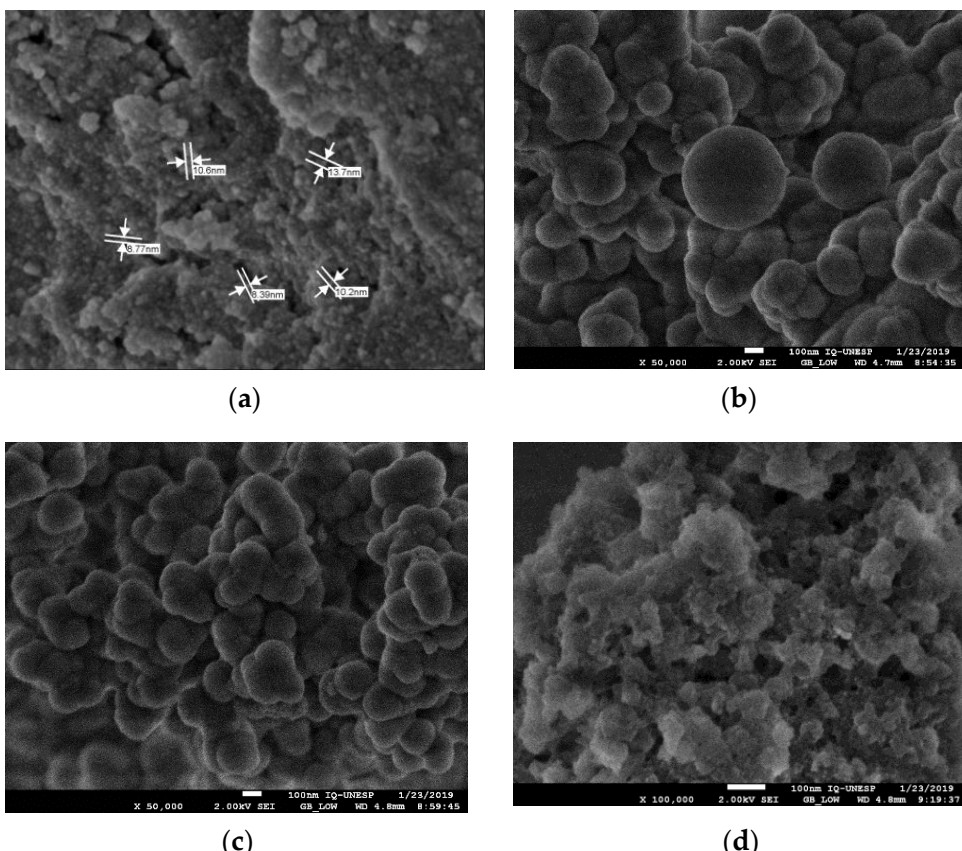

**Figure 5.** SEM Images of the synthesized nanomaterials: (**a**) $Fe_3O_4$—NPS; (**b**) $Fe_3O_4@SiO_2$; (**c**) $Fe_3O_4@MPS$; (**d**) SP-MIP.

The FTIR spectrum (Figure 6) of $Fe_3O_4$-NPS shows two characteristic peaks at 590 $cm^{-1}$ and 1075 $cm^{-1}$, related to Fe-O stretching vibration in tetrahedric sites. The bands near 3200 $cm^{-1}$ and 1600 $cm^{-1}$ refer to the O-H stretching vibration and FTIR spectrum of SP-MIP and SP-NIP) indicating that the surface after polymerization has changed since a wide and broad peak appears between 3100 and 3500 $cm^{-1}$ corresponding to the N-H

bonds of a primary amide of the polyacrylamide formed. The peak is better defined in 1637 cm$^{-1}$ of the vibration of C=O. The weak peaks at 1120 and 1213 cm$^{-1}$ belong to the C-N bands.

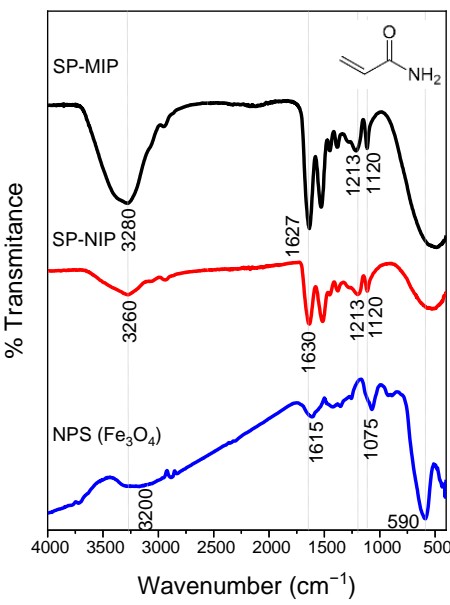

**Figure 6.** FTIR spectra NIP.

### 3.3. HPLC-UV Analysis of AMX

The retention time for AMX was around 1.7 min under the conditions described in Figure 7a. The calibration curve for AMX exhibited a linear range of 5–30 mg L$^{-1}$ with R$^2$ of 0.9996; the limits of detection (LOD) and quantification (LOQ), defined as LOD = 3 × SD/slope and LOQ = 10 × SD/slope, obtained from the calibration curve were 0.147 and 0.491 mg L$^{-1}$, respectively (Figure 7b) [49]. For the LOD and LOQ equations, SD (lowest concentration) is the standard deviation of the (above) baseline noise, which is typically three times the noise level; and LOQ is defined as the lowest amount of analysis that can be reproducibly quantified above the baseline noise [50].

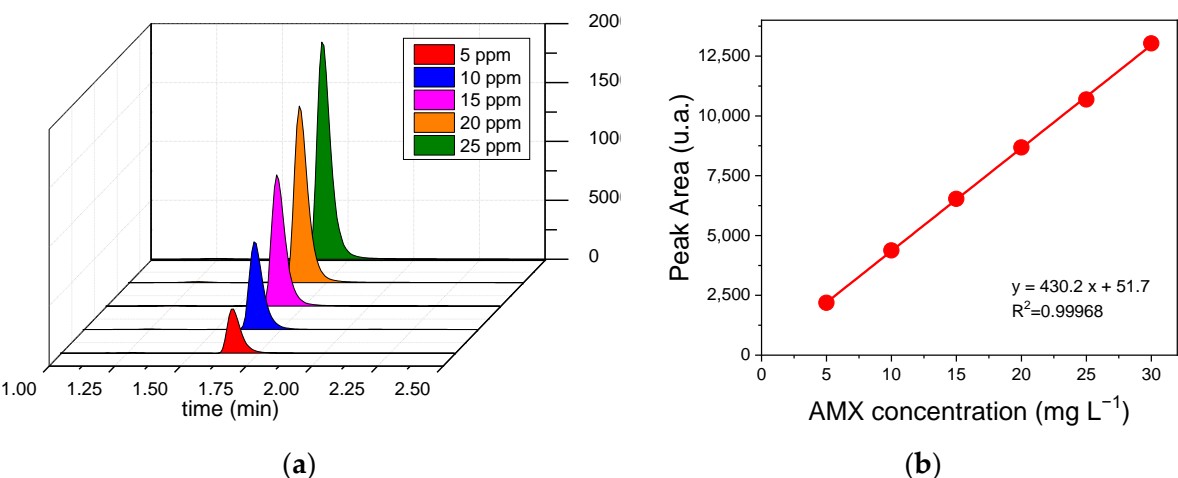

(**a**)      (**b**)

**Figure 7.** Results obtained from the HPLC analysis of Amoxicillin (AMX). (**a**) Chromatogram obtained for Amoxicillin (AMX) (5, 10, 15, 20, and 25 ppm); (**b**) Calibration curve of Amoxicillin (AMX).

### 3.4. Optimizaion of SP-MIP Adsorption

Based on the structure of AMX, in Figure 8, one can predict the occurrence of interactions, such as hydrogen bonding with the adsorbent, leading to the adsorption and

subsequent removal of AMX from the solution. AMX can be found in different forms: when the molecule accepts a proton, forming a cation (AMX$^+$) at a pH level lower than 2.68; in molecular form (AMX) at pH between 2.68 and 7.49 (p$K_{a2}$ = 7.49) in aqueous solution; and in anionic form (AMX$^{1-}$) at pH between 7.49 and 9.63 (p$K_{a3}$ = 9.63). Additionally, AMX exhibits a higher negative charge (AMX$^{2-}$) in solutions at pH values above 9.63 when the molecule loses one or two protons [51,52] (Figure 7a).

**Figure 8.** Amoxicillin (AMX) Ionized forms with different pKa values.

The amount of MIP influences the adsorption of AMX, as shown in Figure 9a. The value of Qe increases rapidly with the increase of SP-MIP up to 6 mg, and higher doses of SP-MIP are found to decrease the capacity of the adsorption of AMX; this may be due to the agglomeration of the adsorbent in the aqueous solution when it is present in high doses. With the application of 6 mg of polymers, one notices the expected differences in terms of the behavior of the MIP and SP-MIP and the NIP and SP-NIP (Figure 9b).

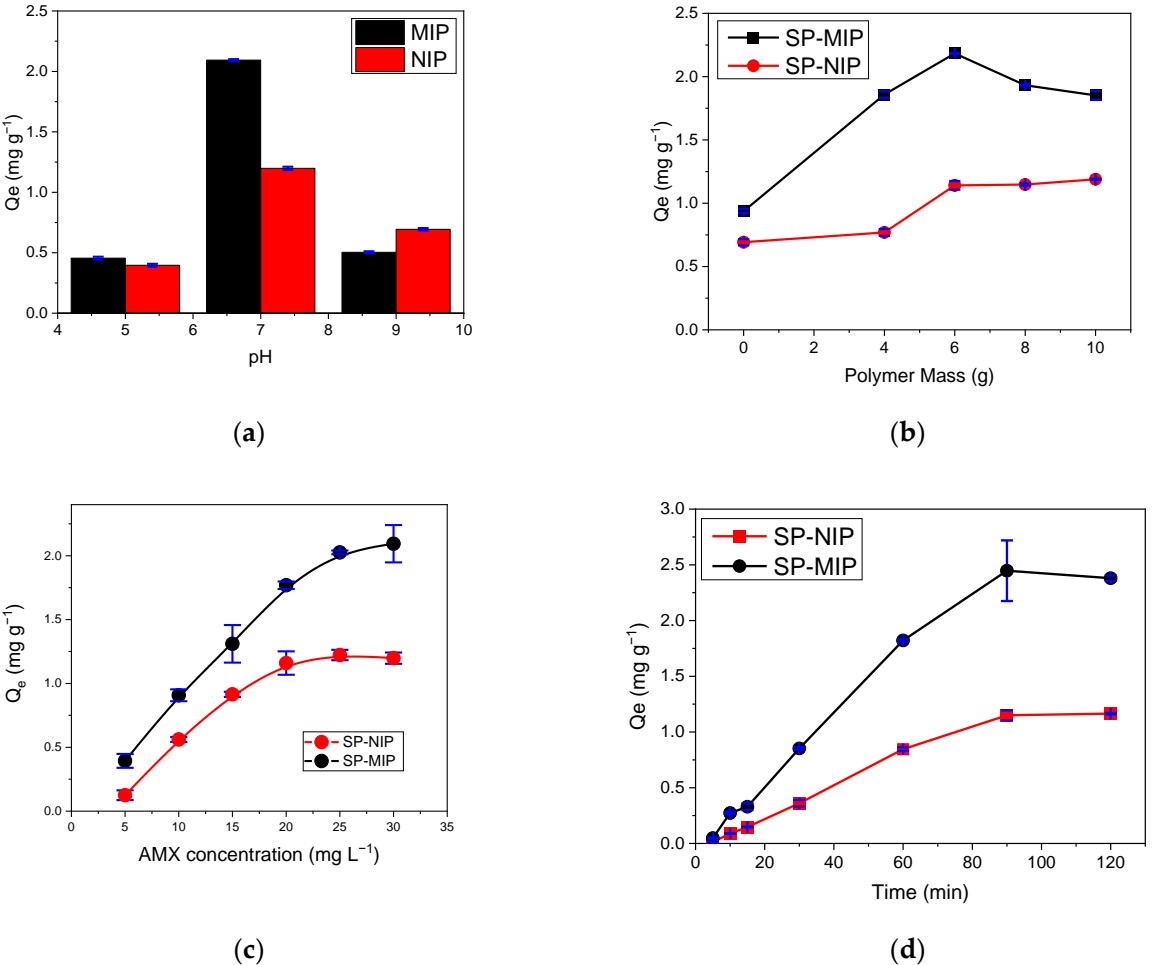

**Figure 9.** Results obtained from the optimization of adsorption on SP-MIP and SP-NIP with (**a**) pH effect, (**b**) polymer mass effect, (**c**) AMX concentration effect, and (**d**) time. All measurements were carried out in triplicate. Table 3 outlines the optimal conditions obtained for the AMX adsorption process.

The binding efficiency of AMX on the SP-MIP and SP-NIP at 25 °C can be found in Figure 9c. It is obvious that as the concentration of AMX increases, the adsorption of AMX on the SP-MIP and SP-NIP increases. The capacity of AMX adsorption of the SP-MIP was found to be higher than that of SP-NIP; this is attributed to the imprinting cavities of the SP-MIP and its high affinity binding sites, which were generated during the cross-linking reactions. The results obtained from the adsorption analysis show that an increase in the AMX concentration from 5 to 30 ppm (10 mg SP-MIP/2 mL) leads to an increase in the adsorption capacity, which is attributed to the availability of active sites during the adsorption process. At the concentration level of 30 ppm, there is a significant change in slope, which indicates a change in the adsorption trend due to a possible saturation (Figure 9c). At this point, one notices the expected differences between the behavior of the SP-MIP and the SP-NIP.

**Table 3.** Optimal conditions for the Amoxicillin (AMX) adsorption process.

| Parameter | Value |
|---|---|
| pH | 7.0 |
| Polymer mass | 6.0 mg |
| AMX concentration | 10 mg L$^{-1}$ |
| Adsorption time | 90 min |

### 3.5. Adsorption Isotherms

Figure 9d presents the adsorption kinetic curves of the SP-MIP and SP-NIP in the presence of 10 mg L$^{-1}$ AMX solution, at pH 7, at room temperature; as can be observed, the adsorption capacity of the polymeric materials increases over time up to 90 min, after which it reaches an equilibrium. The adsorption capacity of SP-MIP was approximately twice that of SP-NIP; this can be attributed to the greater surface area exhibited by SP-MIP, which is attributed to the cavities formed when the polymeric material was subjected to molecular imprinting during the synthesis.

The results obtained from the adsorption process were plotted using three isotherm models: Langmuir, Freunlich, and SIPS; the parameters calculated in each case are shown in Table 4. As can be observed, the SIPS model exhibited a better correlation ($R^2$ = 0.989 for SP-MIP and 0.999 for SP-NIP); this model combines the Langmuir model and the Freundlich model. Based on the result, it can be concluded that our materials have binding site heterogeneity, assumed to predict the heterogeneity of the adsorption systems and limitations associated with the increased concentrations of the adsorbate of the Freundlich model (10,21). The constant $\beta$ values obtained for SP-MIP and SP-NIP were 1.73 and 2.91, respectively; this shows that the system exhibits heterogeneity values greater than 1.00.

**Table 4.** Experimental results were based on Linear fitting for the Langmuir, Freundlich, and SIPS isotherm models.

| | $Q_{max}$ (mg g$^{-1}$) | SP-MIP | SP-NIP |
|---|---|---|---|
| Langmuir [a] | $K_a$ (L mg$^{-1}$) | 12.61 ± (0.0618) | 13.42 ± (0.059) |
| | $R^2$ | 8.37 × 10$^{-3}$ ± (1.03) | 7.87 × 10$^{-3}$ ± (1.08) |
| | $Q_{max}$ (mg g$^{-1}$) | 0.114 | 0.104 |
| Freundlich [b] | $K_a$ (mg$^{1-\beta}$ L$^\beta$ g$^{-1}$) | 0.107 ± (0.091) | 0.0243 ± (0.276) |
| | $\beta$ | 0.95 ± (0.079) | 1.27 ± (0.235) |
| | $R^2$ | 0.960 | 0.850 |
| SIPS [c] | $Q_{max}$ (mg g$^{-1}$) | 2.76 ± (0.36) | 1.35 ± (0.035) |
| | $K_a$ (L μmol$^{-1}$) | 0.0118 ± (0.0055) | 0.0011 ± (3.98 × 10$^{-4}$) |
| | $\beta$ | 1.73 ± (0.28) | 2.91 ± (0.175) |
| | $R^2$ | 0.989 | 0.999 |

*3.6. Selectivity*

One of the key aspects of the SP-MIP is that the selectivity is directly attributed to the binding sites obtained during the synthesis. The results obtained from the selectivity study are shown in Figure 10; the parameters used for the quantitative determination of the selectivity of the proposed SP-MIP are presented in Table 5.

The first thing that can be highlighted is that in the SP-MIP, the imprinting factor ($I_{factor}$) was, in all cases, higher for the AMX, reflecting the improvement in selectivity derived from molecular imprinting. Furthermore, this outcome (higher $I_{factor}$) also shows that the proposed SP-MIP exhibits specific recognition properties that make it highly selective toward AMX in the presence of various possible interferents.

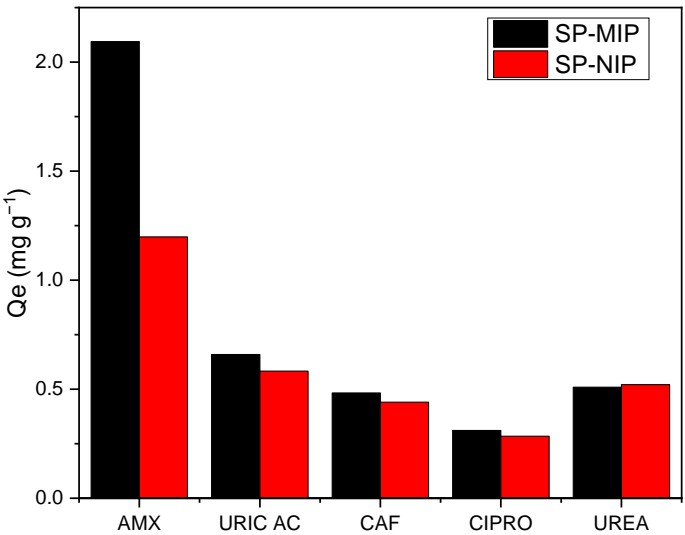

**Figure 10.** Profiles of the results obtained from the selectivity analysis conducted in this study using optimal conditions.

**Table 5.** Selectivity parameters for amoxicillin removal (SP-MIP and SP-NIP) using optimal conditions.

| Interferent | $K_d$-SP-MIP (mL g$^{-1}$) | $K_d$-SP-NIP (mL g$^{-1}$) | S | $I_{factor}$ | $K_{SR}$ |
|---|---|---|---|---|---|
| Amoxicillin | 69.79 | 11.97 | - | 5.83 | - |
| Uric acid | 22.983 | 5.33 | 3.04 | 4.31 | 1.35 |
| Caffeine | 16.094 | 14.674 | 4.34 | 1.10 | 5.3 |
| Ciprofloxacin | 10.360 | 9.483 | 6.74 | 1.09 | 5.35 |
| Urea | 16.970 | 17.372 | 4.11 | 0.97 | 6.01 |

*3.7. Real Sample*

The SP-MIP was used to determine the amount of AMX adsorbed from real samples, including tap water and river water fortified with 30 ppm of AMX under the optimum conditions evaluated in the previous tests conducted in this study. The analysis in real samples was performed in triplicate, and the AMX present in the samples was calculated using the calibration curve. Table 6 shows the recovery percentages obtained in the quantification of AMX, which were 94.3% on average; this result shows that the synthesized material is a suitable and efficient alternative tool for extracting AMX in these matrices without the need for pretreatment.

**Table 6.** % Recovery from superparamagnetic Imprinted polymer (SP-MIP) in the analysis of regional river and tap water and two types of amoxicillin pills. Standard deviation for n = 3.

| Samples | AMX Added (ppm) | % Recovery |
|---|---|---|
| Tap water (Laboratory, Lima, Perú) | 30 | 94.43 ± 0.13 |
| River Water (Puquio, Huancayo, Perú) | 30 | 92.82 ± 0.14 |
| Amoxicillin pills (generic formulation) | 30 | 95.30 ± 0.02 |
| Amoxicillin pills (Commercial formulation) | 30 | 94.99 ± 0.13 |

## 4. Conclusions

The present study reported the development of magnetite superparamagnetic nanoparticles (SNP) and the synthesis of superparamagnetic molecularly imprinted polymer (SP-MIP) with a core-shell structure for the selective detection of AMX in real samples. Compared with the non-imprinted polymer and other components from the literature, as shown in Table 7, the SP-MIP demonstrated a significantly higher capacity and LOD for AMX. Selectivity adsorption tests demonstrated that SP-MIP is highly selective for AMX in the presence of other interfering molecules. The results obtained from applying SP-MIP using HPLC for the quantification of AMX in samples of tap water, river water, and drugs pointed to an average recovery percentage of around 94.3%. The proposed mag-MIP extraction method exhibited simplicity, rapidity, and satisfactory selectivity. Based on the findings of this study, one can conclude that the proposed SP-MIP is a novel efficient alternative tool that can be used for the effective determination of AMX in aqueous matrices.

**Table 7.** Comparative study of the fabricated material with the literature for Amoxicillin (Amx).

| Material | Analyte/ Real Sample | LOD/ % Recovery | Ref. |
|---|---|---|---|
| MIP coated on CdTe quantum dots | Amoxicillin/ egg, milk, and honey | $0.14 \ \mu g \ L^{-1}$/ 85–102% | [53] |
| MIP grown on MWCNT surface | Amoxicillin/ milk and honey | $8.9 \times 10^{-10} \ mol \ L^{-1}$/ 88–96% | [54] |
| Magnetic MIP—CPE Sensor | Amoxicillin/ Capsule | $0.26 \times 10^{-9} \ mol \ L^{-1}$/ 98.8 and 103.2% | [55] |
| Magnetic MIP—CPE Sensor | Amoxicillin/ milk and river water | $0.75 \times 10^{-6} \ mol \ L^{-1}$/ 96–100% | [8] |
| Hybrid MIP | Amoxicillin/ Tap water | $73 \times 10^{-12} \ mol \ L^{-1}$/ 93–96% | [56] |
| This work | Amoxicillin/river water, tap water, and pills | $0.147 \ mg \ L^{-1}$ or $4.02 \times 10^{-7} \ mol \ L^{-1}$/93–95% | |

**Author Contributions:** Methodology: R.L., S.K. and S.E.T.; Writing-Review and Editing: R.L., S.K., A.W., M.D.P.T.S. and G.P.; Supervision, S.K., M.D.P.T.S. and G.P. All authors have read and agreed to the published version of the manuscript.

**Funding:** This research was funded by PROCIENCIA, PE501080434-2022-PROCIENCIA, project n° 067-PROCIENCIA-2022, and by CNPq-Brazil (301728/2019-4).

**Institutional Review Board Statement:** Not applicable.

**Informed Consent Statement:** Not applicable.

**Data Availability Statement:** Not applicable.

**Acknowledgments:** The authors thank all the researchers involved in this project. The authors wish to express their gratitude to PROCIENCIA for the financial assistance granted to the projects: PE501080434-2022-PROCIENCIA, 067-PROCIENCIA-2022 and CNPq-Brazil (301728/2019-4) in support of this research.

**Conflicts of Interest:** The authors declare no conflict of interest.

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
