# Peer review of "Synthesis and Characterization of Magnetic Molecularly Imprinted Polymer for the Monitoring of Amoxicillin in Real Samples Using the Chromatographic Method"

_magnetochemistry, doi:10.3390/magnetochemistry9040092_

Round 1

Reviewer 1 Report

This is an interesting work dealing with the synthesis and characterisation of the molecularly imprinted superparamagnetic polymer for amoxicillin removal. Having in mind the intention of the work, it is necessary to address the adsorption methods that are often incorrectly presented in adsorption studies. 

  1. Particular attention should be paid to the fact that a linearized model of a nonlinear function should not be drawn in such a way that Qe is found on both sides of the equation because this artificially raises the coefficient of correlation [J.-Z. Zhang, Avoiding spurious correlation in the analysis of chemical kinetic data, Chem. Commun., 2011, 47, 6861, DOI: 10.1039/c1cc11278c;E. D. Revellame, D. L. Fortela, W. Sharp, R. Hernandez and M. E. Zappi, Adsorption kinetic modelling using pseudo-first-order and pseudo-second-order rate laws: A review, Clean. Eng. Technol., 2020, 1, 100032, DOI: 10.1016/J.CLET.2020.100032]. 
  2. Ho and McKay’s work [Process Biochemistry 34 (1999) 451] concluded that in all adsorption systems that they analyzed, pseudo-second-order (PSO) kinetics provided the best correlation with experimental data. This was quickly adopted by researchers in this field, and nearly 90% fitted the PSO model for most of the adsorption systems in the recent literature, according to a review by Revellame. Some studies did not consider pseudo-first-order (PFO) at all, with only 9% nonlinear modelling. Linearization of a non-linear model is a problem as it boosts correlation coefficients. In the linear PFO forms, the left equation side becomes undefined as soon as the equilibrium capacity has been reached. On the other hand, all linear PSO forms are applicable for any t > 0. As it is more accurate to use non-linear modelling, along with the fact that most systems reach equilibrium adsorption fast, only the first steps of the adsorption can be accurately modelled. Especially, when there is no dissociation of adsorbate, PFO is the only logical choice. 
  3. Is there perspective propositions in the literature on how to deal with saturated adsorbent waste disposal/reuse?
  4. Please include error bars in the recovery values in Table 6. Moreover, each measurement result needs to be presented with a standard deviation, this is missing all results.
  5. What is an environmentally significant AMX amount in water? Adsorption capacities are quite low, please include a literature comparison with similar materials.

This is a well-defined study with comprehensive insight SP-MIP, therefore I recommend major revision after the necessary corrections are made.

Author Response

Dear Reviewer

Reviewer 2 Report

The authors present a synthesis of superparamagnetic molecularly imprinted polymer (SP-MIP) with a core-shell structure for the selective detection of AMX in real samples. In comparison with the non-imprinted polymer (SP-NIP), the SP-MIP  they seem to demonstrate a higher capacity for AMX binding.

The results seem to be sustained by the data so the overall opinion is good but there are some aspects that need to be clarified before accepting it for publication:

-In Fig.7, in the plots used for the adsorption optimization there are error bars of Qe only for AMX concentration dependency and the optimal parameters values are given without a trust range. The authors should explain how they calculated the errors in fig 7 c) and why this does not apply for the other plots and for the results given in Table 3.

- In Table 5 and in Fig 8 there is a comparison with other substances but there are no information about their source (other measurements, other prepared samples or the literature). They are meant to prove the  higher absorption rate for amoxicillin but there are no details about procedures and conditions.

Author Response

Dear Reviewer

Reviewer 3 Report

The research article entitled: Synthesis and characterization of magnetic molecularly imprinted polymer for the monitoring of Amoxicillin in real samples using the chromatographic method present an interesting topic and is of good quality. Nevertheless, some major revision is necessary. The following issues should be addressed:

1. All equations in chapter 2 are looking blurry, like they are images. Please rewrite all equations with a suitable equation writing editor.

2. Chapter 2.1: Chemicals were purchased from Sigma Aldrich and Merck and the reader can guess what is from which manufacturer. Sorry, this needs to be corrected. Chemicals should be metioned as follows: (abbreviation/molecular formula, grade/molecular weight for polymers, manufacturing company, city of manufacturing, country of manufacturing

3. Chapter 2: The used devices are not mentioned correctly, even with which device the ultrapure water was produced. For the devices: (type, manufacturer, city of manufacturing, country of manufacturing).

4. Chapter 2.2: The used software should mentioned correctly (version, developer, city, country).

5. In the introduction, the biomedical application of magnetite nanoparticles should be mentioned. The following papers should be cited: https://doi.org/10.1016/j.jcis.2019.01.103  https://doi.org/10.3390/magnetochemistry6040068

6. In the introduction, the fabrication methods for magnetite nanoparticles should be mentioned, as the authors used the heating method, the wet-chemical method as applied here is missing in the introduction.

7. The authors should provide some microscopic images of the synthesized nanoparticles.

8. What is the size of the synthesized particles and their size distribution? This is missing and should be provided. 

9. What about zeta potential and hydrodynamic diameter of the synthesized particles with and without AMX at different concentrations absorbed on the nanoparticle surfaces in ultrapure water, tap water and river water? It should be useful for understanding the characterization of the particles and their monitoring in solutions, as the absorption isotherms are giving quantitative information. Somehow the quality is missing in this study. Please provide this investigation.

10. The analytic verification of the nanoparticles produces is missing. Also the analytic proving of the AMX absorption on the particle surfaces is missing. Please provide suitable analytic measurements. For example, FTIR or UV-VIS. 

11. The figure and table captions are too uninformative and should be rewritten. In order to make them easier understandable, the abbreviations should be defined also in the captions. Moreover, the captions should help to understand the figures and tables better. Especially for the figures, the investigation should be clearer communicated and written what is there to seen. Please improve them.

12. Figure 2: The unit in the y-axis is written in a different style as in the following figures (units in brackets). Please standardize that.

13. Figure 7: The font size of the diagram axes are too small for the numbers and the axis labels. Please correct this.

14. Sometimes space signs are missing between a value and a unit. For example in the figure 5 caption. Please fix that throughout the whole manuscript.

15. It looks like that not the correct degree sign (Alt + 0176) is used in this manuscript. Please correct this in the manuscript.

16. The conclusion should precise in which way the results (outlook) can be used in further studies or applied for others. Please expand the conclusion in this point.

Author Response

Dear Reviewer

Reviewer 4 Report

Manuscript Title: Synthesis and characterization of magnetic molecularly im- printed polymer for the monitoring of Amoxicillin in real sam- ples using the chromatographic method

Journal Title: Magnetochemistry

Authors: Rosario López, Sabir Khan, Ademar Wong, Maria D.P.T Sotomayor, and Gino Picasso

Manuscript ID: magnetochemistry-2284544

The article can be published in Magnetochemistry after major revision. Authors should consider the comments presented below:

- Figure 2. Have the authors tried to verify in practice the results of semi-empirical computational simulations. I mean, have you tried using the lowest binding energy monomer to bind amoxicillin, or maybe make a reference to a published work to verify obtained semi-empirical computational simulations. Is it correct that you extend the data for the monomer then to the resulting polymer? Please comment it. Does this take into account non-specific interactions between the monomer and the target compound?

- Figure 6. Please make a short description why R2NH group is not in ionized state (like R2NH2+) according to the scheme. 

- Figure 7a, Figure 7b and Figure 7d, do not have error bars (or error bars are too small?). Please add error bars.

- Please add a comparative analysis of the obtained adsorbent with other available adsorbents to show perspective of use from practical point of view and in the efficiency of determining of Amoxicillin.

Author Response

Dear Reviewer

Round 2

Reviewer 1 Report

The authors made necessary corrections in the revised manuscript, therefore I recommend acceptance.

Reviewer 2 Report

My decision is: Accept in present form

Reviewer 3 Report

The authors of the manuscript entitled Synthesis and characterization of magnetic molecularly imprinted polymer for the monitoring of Amoxicillin in real samples using the chromatographic method improved the quality of the manuscript significantly. Therefore, I suggest the editor to accept this paper for publication.

Reviewer 4 Report

The authors have addressed all my comments/concerns in this revised manuscript. I have no additional concerns regarding this manuscript.

Manuscript could be recommended for publication.